# Natriuretic-like Peptide Lebetin 2 Mediates M2 Macrophage Polarization in LPS-Activated RAW264.7 Cells in an IL-10-Dependent Manner

**DOI:** 10.3390/toxins15040298

**Published:** 2023-04-19

**Authors:** Dorsaf Bouzazi, Wael Mami, Amor Mosbah, Naziha Marrakchi, Melika Ben Ahmed, Erij Messadi

**Affiliations:** 1Plateforme de Physiologie et Physiopathologie Cardiovasculaires (P2C), Laboratoire des Biomolécules, Venins et Applications Théranostiques (LR20IPT01), Institut Pasteur de Tunis, Université Tunis El Manar, Tunis 1068, Tunisia; dorsaf.bouzazi@pasteur.utm.tn (D.B.);; 2Laboratory of Biotechnology and Bio-Geo Resources Valorization (LR11ES31), Higher Institute of Biotechnology of Sidi Thabet (ISBST), University of Manouba, Tunis 2010, Tunisia; 3Laboratoire de Transmission, Department of Clinical Immunology, Contrôle et Immunobiologie des Infections, Institut Pasteur de Tunis, Université Tunis El Manar, Tunis 1068, Tunisia

**Keywords:** snake venom-derived natriuretic peptide, lebetin 2, inflammation, cytokines, macrophage polarization, interleukin-10, cardioprotection

## Abstract

Snake natriuretic peptide (NP) Lebetin 2 (L2) has been shown to improve cardiac function and reduce fibrosis as well as inflammation by promoting M2-type macrophages in a reperfused myocardial infarction (MI) model. However, the inflammatory mechanism of L2 remains unclear. Therefore, we investigated the effect of L2 on macrophage polarization in lipopolysaccharide (LPS)-activated RAW264.7 cells in vitro and explored the associated underlying mechanisms. TNF-α, IL-6 and IL-10 levels were assessed using an ELISA assay, and M2 macrophage polarization was determined by flow cytometry. L2 was used at non-cytotoxic concentrations determined by a preliminary MTT cell viability assay, and compared to B-type natriuretic peptide (BNP). In LPS-activated cells, both peptides reduced TNF-α and IL-6 release compared to controls. However, only L2 increased IL-10 release in a sustained manner and promoted downstream M2 macrophage polarization. Pretreatment of LPS-activated RAW264.7 cells with the selective NP receptor (NPR) antagonist isatin abolished both IL-10 and M2-like macrophage potentiation provided by L2. In addition, cell pretreatment with the IL-10 inhibitor suppressed L2-induced M2 macrophage polarization. We conclude that L2 exerts an anti-inflammatory response to LPS by regulating the release of inflammatory cytokines via stimulating of NP receptors and promoting M2 macrophage polarization through activation of IL-10 signaling.

## 1. Introduction

Myocardial infarction (MI) triggers an intense inflammatory response that is essential for cardiac repair, but which is also involved in the pathogenesis of post-MI remodeling and heart failure (HF) [1,2]. In experimental studies, specific approaches targeting inflammatory mediators were successful in attenuating ischemic injury, leading to considerable enthusiasm regarding their potential in MI patients. They have been proven to markedly reduce infarct size (IS) in reperfused MI and prevent the extension of necrosis and subsequent fibrosis, which is a major determinant of impaired cardiac function after MI [1,2]. Unfortunately, despite promising data from animal studies, translation of inflammation treatment into therapy for MI patients was unsuccessful and several anti-inflammatory approaches failed to reduce IS in clinical investigations [3].

B-type natriuretic peptide (BNP), a member of the natriuretic peptide (NPs) family that also includes A-type (ANP), C-type (CNP) and snake [4] natriuretic peptides, has been demonstrated to be a potent cardioprotective molecule in MI [1,5,6,7,8,9,10,11,12] and suggested to have anti-inflammatory properties. Several studies have shown that BNP and its receptors are expressed and differentially regulated in inflammatory cells [13,14], and that there is an association between its cardioprotective actions in MI and inflammation [1,15]. Nevertheless, few reports have addressed the role of this peptide in the regulation of inflammatory mediators and macrophage polarization after MI [1,16]. So far, the evaluation of the underlying inflammatory mechanisms of BNP is still controversial and not thoroughly understood [17,18]. In previous study, the snake natriuretic-like peptide Lebetin 2 (L2) [19,20] was shown to modulate post-MI inflammation by reducing inflammatory cell recruitment and promoting macrophage polarization in the infarct area of MI rats [1]. Besides its inflammatory effects, L2 has also been found to exert potent cardioprotective actions against myocardial ischemia–reperfusion (IR) injury in both acute and late stages of MI by decreasing necrosis and fibrosis, enhancing regeneration of endothelial cells and cardiomyocytes, and improving left ventricular function and coronary flow [1,12]. L2 cardiac effects have been demonstrated to be mediated through a BNP-like mechanism, involving the NP receptor A (NPR-A)/cyclic guanosine monophosphate (cGMP)-mediated pathway [12]. However, major discrepancies have been reported between the two peptides [1,12,21], particularly in relation to macrophage polarization. In this context, only L2 has been shown to promote M2-like macrophages, while BNP failed to elicit the polarization of macrophages after MI [1].

Macrophage polarization plays a central role in the development and resolution of the inflammatory response to MI. In particular, the switch from the classically activated M1-like phenotype in the pro-inflammatory phase to the alternatively activated M2-like macrophages in the reparative phase has been shown to prevent post-MI adverse left ventricular remodeling and fibrosis in the injured heart [22,23,24,25,26,27]. The mechanism of switching from one phenotype to another could involve the pro-inflammatory biomarkers secreted by the M1-type, i.e., tumor necrosis factor alpha (TNF-α) and interleukin-6 (IL-6) cytokines, and the anti-inflammatory biomarkers secreted by the M2-type, i.e., transforming growth factor beta (TGF-β) and interleukin (IL-10) [28,29,30]. An increasing number of reports indicate that the polarization mechanism would be driven by IL-10 as a major determinant, and requires sustained IL-10 signaling to maintain this phenotype [31,32]. Perturbations affecting both balance and transition between pro- and anti-inflammatory phases can exacerbate acute MI and thus aggravate post-infarction clinical outcomes [28,29,30]. Therefore, pharmacological modulation of macrophage polarization into their anti-inflammatory phenotype may provide a therapeutic strategy for enhancing the reparative phase following MI.

In this context, we sought to investigate the effect of L2 on macrophage polarization in vitro on lipopolysaccharide (LPS)-activated RAW264.7 macrophage cells and to explore the associated underlying mechanisms. Therefore, M2 macrophage polarization was assessed as well as the pro-inflammatory M1-type (TNF-α and IL-6) and anti-inflammatory M2-type (IL-10) cytokine levels. The involvement of the NPR-A-cGMP pathway and IL-10 in the observed effects was also determined. The main data showed that L2 regulates the inflammatory response in LPS-stimulated RAW264.7 cells by decreasing pro-inflammatory M1 macrophages and triggering M2 macrophage polarization, as well as reducing the release of pro-inflammatory cytokines and increasing the secretion of the anti-inflammatory cytokine IL-10. Our results further demonstrated that L2 uses the NPR-A/cGMP/IL-10 axis to promote M2 macrophage-like phenotype. Our findings suggest a previously undiscovered mechanism underlying L2-mediated inflammatory effects and provide a new insight that L2 may be of a great therapeutic significance the enhancement of the reparative phase after MI, especially in patients with intense and prolonged inflammatory response.

## 2. Results

### 2.1. Effects of L2 and BNP on RAW264.7 Cell Viability

To investigate the effects of L2 and BNP on cell viability and determine the best concentrations for further inflammation studies, MTT assay was performed 24 h after addition of the molecules (Figure 1). After ensuring the RAW264.7 cell response to H_2_O_2_- and Triton-induced cytotoxicity, cell viability was observed to be more than 90% after L2 and BNP treatments at all concentrations. L2 and BNP induced a maximum viability decrease of 9% and 10%, respectively, at concentrations of 0.8 ng/mL and 3.2 ng/mL, suggesting that both L2 and BNP have a weak toxicity to RAW264.7 cells. In line with this result, concentrations ranging from 0.05 to 6.4 ng/mL were selected to be tested in the subsequent studies.

### 2.2. Effects of L2 and BNP on LPS-Induced Inflammatory Cytokines

RAW264.7 cells inflammatory response was induced by LPS, and effects of L2 and BNP on the inflammatory response were studied by evaluating the protein levels of TNF-α, IL-6 and IL-10 using ELISA assay. Results showed that in both experimental conditions (Figure 2 and Figure 3), when treated with LPS, levels of TNF-α, IL-6 and IL-10 significantly increased compared to the control LPS untreated cells (*p* < 0.001).

L2 treatment reversed the above effects induced by LPS, with a maximum decrease of TNF-α and IL-6 levels at 0.05 (−45%, *p* < 0.001) and 0.2 ng/mL (−71%, *p* < 0.001), respectively (Figure 2a,b). L2 also significantly increased the level of IL-10 under all the concentrations ranging from 0.05 to 3.2 ng/mL, with a maximum increase at the concentration of 0.1 ng/mL (77%, *p* < 0.001) (Figure 2c).

When cells were pretreated with BNP, TNF-α and IL-6 were significantly reduced, with a maximum decrease at the concentration of 0.2 ng/mL (−40%, *p* < 0.05) and 1.6 ng/mL (−48%, *p* < 0.05) for TNF-α and IL-6, respectively (Figure 3a,b). BNP treatment increased the level of IL-10 to a significant extent at the concentration of 1.6 ng/mL (23%, *p* < 0.01) (Figure 3c).

TNF-α/IL-10 ratio markedly decreased after L2 treatment (Figure 2d, *p* < 0.01) but to a lesser extent after BNP (Figure 3d). These results indicate that L2 and BNP may modulate the production of inflammatory factors in a different way.

To investigate if the L2 and BNP inflammatory effects were mediated through natriuretic receptors, cells were pretreated with the NP receptor antagonist isatin before L2 and BNP treatments. When administered alone, isatin had no statistically significant effect on inflammatory cytokine levels (Figure 2 and Figure 3). However, NP receptor blockade by isatin abolished the inhibitory effect of L2 and BNP on TNF-α and IL-6, suggesting that regulation of these cytokines is mediated by natriuretic receptors (Figure 2 and Figure 3). Regarding IL-10, isatin suppressed only the effect of L2 on IL-10 increase, but not of BNP (Figure 2c and Figure 3c).

### 2.3. Effects of L2 and BNP on Macrophage Polarization and Related Mechanisms

To investigate the effect of L2 and BNP on macrophage polarization and its underlying mechanism, flow cytometry analysis was used to detect expression levels of total macrophage marker CD68 and M2-type macrophage marker CD206 in RAW264.7 macrophages. To detect M1-like versus alternatively activated macrophages, the M2 macrophage polarization was determined by counting CD68 and CD206/MRC-1 double-labeled cells. Only L2, not BNP, significantly decreased the number of M1-like macrophages compared to the untreated control group, with a significant reduction at the concentration of 0.8 ng/mL (−43%, *p* < 0.001) (Figure 4b,d). Similarly, only L2 treatment, and not BNP, significantly increased the level of CD206 expression with a maximum increase at the concentration of 0.4 ng/mL (212%, *p* < 0.001) compared to the control group challenged only with LPS (*p* < 0.001) (Figure 4c,e).

The blockade of natriuretic receptors with isatin did not affect the level of M1 and M2 macrophages (Figure 4b,c); however, it completely abolished the M1/M2 regulatory effect of L2 compared to the group treated only with L2 (Figure 4b,c).

On the other hand, treatment of LPS-activated RAW264.7 cells with IL-10 reproduced the effect of L2 by increasing the polarization of M2 macrophages in a dose-dependent manner (Figure 5b). However, cell pretreatment with IL-10 inhibitor completely abolished the effect of L2 on CD206 expression level compared to the group treated with L2 alone (*p* < 0.001) (Figure 5d), thus suggesting that IL-10 signaling is the primary determinant of L2-induced M2 macrophage polarization in LPS-activated RAW264.7 cells. Exogenous IL-10 did not affect the expression of M1-like macrophages (Figure 5a), whereas cell pretreatment with the IL-10 inhibitor abolished the L2-induced M1 decrease (Figure 5c).

## 3. Discussion

Acute MI, one of the most lethal diseases globally [33,34], triggers an intense inflammatory response that promotes cardiac dysfunction, cell death and subsequent ventricular remodeling [2]. Currently, there are no clinically applicable anti-inflammatory medications for myocardial ischemic injury [3]. In the cardiovascular system, the natriuretic-like peptide L2 has been reported to play a cardioprotective role in MI by modulating the post-ischemic inflammatory response [1,12]. Nevertheless, a deeper understanding of these effects is still missing and so far there have been few mechanistic studies on NP and their role in the regulation of macrophage subtypes after MI [1].

In the present study, we assessed the effects of L2 on the inflammatory response in LPS-activated macrophage RAW 264.7 cells in vitro and compared them to the effects of BNP. The in vitro model using RAW 264.7 cells has been widely used and approved to promote new anti-inflammatory drugs [35]. RAW 264.7 cells have been demonstrated to express all three NP receptors (NPR-A, NPR-B and NPR-C) [13,14] and have the plasticity to switch from pro-inflammatory M1-like macrophages to the anti-inflammatory M2-like phenotype. As expected, in our experiments, LPS treatment markedly increased the release of pro-inflammatory cytokines TNF-α and IL-6 as previously reported in RAW264.7 cells [36]. We also found that LPS increased the secretion of the anti-inflammatory cytokine IL-10. In the literature, LPS treatment has been reported to induce a biphasic increase in IL-10 levels in vivo with an early IL-10 increase through dynamic spatio-temporal regulation involving LPS-associated inflammatory mediators [30]. Thus, studying the effects of L2 and BNP on inflammatory alteration in LPS-activated cells may provide in vitro evidence for the effect of both molecules on inflammatory cytokines as well as macrophage phenotype regulation.

First, we examined the effects of L2 and BNP on cell viability of RAW 264.7 cells using an MTT assay and found that both peptides exhibited very low toxicity, as reported previously [36]. Based on MTT assay results, the appropriate concentrations of L2 and BNP were tested to determine their effects on LPS-induced inflammatory cytokines. Both peptides significantly decreased levels of the pro-inflammatory cytokines TNF-α and IL-6, but only L2 resulted in consistent release and stable increase in IL-10 secretion at all concentrations in LPS-stimulated cells. In previous studies reporting similar results, BNP has been proven to inhibit LPS-induced TNF-α and IL-6 expression, in vitro and in vivo, at both protein and mRNA levels [36,37], as well as in patients with acute MI [38]. However, in the literature there are still some discrepancies regarding the effect of NPs on IL-10 expression. Some studies indicated that BNP can induce IL-10 in both human THP-1 macrophages [39] and murine LPS-activated RAW264.7 cells in a dose-dependent manner [36], whereas other NP family members such as ANP failed to increase IL-10 secretion in LPS-activated cells [40]. In our experimental setting, although BNP induced a moderate increase in IL-10 level, this effect remains unchanged after the pharmacological blockade of natriuretic receptors, which confirms the weak effect of BNP on this cytokine. Conversely, the regulatory effect of L2 on inflammatory cytokines was found to be mediated through activation of the NPR-A/cGMP pathway, as these effects were abolished after selective NPR-A blockade with isatin. How NPR-A-cGMP signaling could modulate inflammatory cytokines is unclear and awaits further investigation. Nevertheless, a study reported that ANP uses a similar cGMP-dependent mechanism to modulate the level of expression of inflammatory mediators such as TNF-α, via a transcriptional process involving NF-κB in LPS-activated macrophages [41].

Our data further showed that L2 decreased the M1 phenotype marker expression and enhanced the M2-like macrophage phenotype when BNP failed to promote macrophage polarization in LPS-activated cells, thus confirming previous findings performed in vivo [1]. Like L2, other NPs, such as CNP, which shares the highest homology and identity with L2 [21], have been reported to induce M2 polarization in vitro [42]. In our study, both NPR-A blockade and IL-10 inhibition completely suppressed the L2-provided M2 macrophage polarization in LPS-activated RAW264.7 cells. This indicates that L2 uses the NPR-A/cGMP/IL-10 axis to promote the M2-like macrophage phenotype, and that IL-10 is the major determinant of L2-induced M2 macrophage polarization. According to the literature, the mechanism of M2 polarization requires sustained IL-10 signaling to maintain this phenotype as IL-10 is rapidly cleared from the injury site [32]. This hypothesis is reinforced by the fact that in our work, BNP, which does not increase IL-10 in a sustained manner, does not trigger M2-like macrophage polarization either, whereas L2 which promotes a constant and stable IL-10 increase at all concentrations also enhances M2 subtype macrophages. Although other anti-inflammatory cytokines such as interleukin-4 (IL-4) may be involved in the conversion of M1- to M2-like macrophages [43], several studies have reported that the IL-10-induced M2-subtype is the most widely dominant [32]. IL-10 has also been pointed to play a central role in mediating the phenotypic conversion from M1 to M2-like macrophages in MI [22,23,24,25,26,27]. In this setting, IL-10 has been shown to decrease IS, necrosis and fibrosis and improve LV function and remodeling in mice after infarction [44,45,46], and also reduce the incidence of HF progression after angioplasty in patients [47]. Several other reports have indicated that IL-10 may exert anti-inflammatory actions in a variety of cell types [48,49,50].

Considerable data have demonstrated that macrophage phenotype transition from M1-like inflammatory to M2-like anti-inflammatory phenotype is important for cardiac healing following MI [22,23,24,25,26,27]. However, the exact mechanism mediating this monocyte phenotypic conversion remains unclear. A dynamic interaction exists between pro- and anti-inflammatory mediators at the level of both inflammatory cells and cytokines generated after MI in an optimized trade-off between cardiac healing and disease aggravation. Therefore, perturbations in the balance and transition between pro-inflammatory and anti-inflammatory reparative phases can exacerbate acute MI and worsen its clinical outcomes [3]. For instance, it has been reported that inhibiting pro-inflammatory cytokines such as TNF-α and IL-6 can stimulate the IL-10/STAT3 signaling pathway and inhibit inflammatory functions of macrophages [28,30], while their activation can, on the other hand, delay the M2 macrophage polarization after MI [29]. Similarly, loss of IL-10 can increase the level of pro-inflammatory cytokines such as IL-6 and exacerbate inflammation in several diseases [51]. In addition to the role of IL-10 in inflammation resolution, it has been reported that M1 regulation is also important in modulating subsequent M2 behavior, thus correcting M1 homeostasis may be a therapeutic target in dysfunctional M2 activation [29,43]. Here, we demonstrated that L2 is able to mediate the balance between pro- and anti-inflammatory mediators at both macrophage and cytokine levels in vitro, by inhibiting M1-like macrophages and related biomarkers (i.e., TNF-α and IL-6) and up-regulating M2-like subtype and related biomarkers (i.e., IL-10). Recently, several clinical trials have been performed and novel therapeutic tools developed with the aim of targeting either IL-10 or IL-6 against MI-induced cardiac damage [3,52], which could correct the imbalance between the pro-and anti-inflammatory phases and thus accelerate cardiac healing. Therefore, by combining both the inhibition of pro-inflammatory mediators and the potentiation of anti-inflammatory mediators, L2 could be a relevant tool to trigger the resolution of the inflammatory process.

## 4. Conclusions

In conclusion, the present study established a better mechanistic understanding of the anti-inflammatory role of L2 in vitro. Combined with previous results [1], our findings support the idea that L2 induces an increased release of IL-10 by activating NPR-A/cGMP signaling, inducing downstream M2 macrophage polarization in turn, which may prevent adverse post-MI fibrosis and left ventricular remodeling and provide cardioprotection. In this work, we further confirmed that L2 exerts stronger anti-inflammatory actions than BNP, which failed to promote M2 macrophage polarization in vitro and in vivo [1]. By combining both natriuretic-mediated actions and anti-inflammatory properties, L2 could be a promising and effective drug for post-MI treatment.

## 5. Materials and Methods

### 5.1. Drugs and Reagents

Synthetic L2 (isoform α, 38 amino acids) was purchased from Genosphère Biotechnologies (Paris, France). BNP (human recombinant peptide, 32 amino acids) was purchased from Sigma–Aldrich (Saint Quentin Fallavier, France). Isatine was purchased from Cliniscience (Nanterre, France). Mouse Interleukin-10 (IL-10) and IL-10 antibody were purchased from R&D Systems (Minneapolis, MN, USA).

### 5.2. Experimental Research Design

M2 macrophage polarization was assessed in vitro by flow cytometry on lipopolysaccharide (LPS)-activated RAW264.7 cells, as well as the pro-inflammatory M1-type (TNF-α and IL-6) and anti-inflammatory M2-type (IL-10) cytokine levels by ELISA assay (Figure 6). The involvement of the NPR-A-cGMP pathway and IL-10 in the observed effects was determined by pretreating the cells with isatin (natriuretic receptor blocker) and the IL-10 antibody, respectively (Figure 6).

### 5.3. Cell Culture

RAW264.7 cells were obtained from a mouse macrophage cell line (ATCC^®^ TIB-71™, Manassas, VA, USA). Briefly, cells were cultured in RPMI Medium 1640 (Thermo Fisher Scientific, Waltham, MA, USA) supplemented with 10% Gibco^®^ fetal bovine serum (Thermo Fisher Scientific, USA) and 100 μg/mL penicillin-streptomycin (Sigma-Aldrich Co., Burlington, MA, USA).

### 5.4. MTT Assay Protocol for Cell Viability

Normal MTT (3-(4,5-dimethylthiazol-2-yl)-2,5-diphenyltetrazolium bromide) assay [53] was used to evaluate the cell viability after L2 and BNP treatment for 24 h to determine their non-cytotoxic concentrations to be used in subsequent inflammation experiments.

Briefly, different groups of cells were seeded (1×10^4^ cells/well) in 96-well plates and were cultured for 24 h. Then, cells were treated with L2 or BNP under different concentrations (0, 0.05, 0.1, 0.2, 0.4, 0.8, 1.6, 3.2, 6.4, 12.8, 25.6, 51.2 and 100 ng/mL) and incubated at 37 °C for 24 h. Then, 25 µL of MTT solution (500 µg/mL) was added to each well and plates were incubated at 37 °C for 4 h. Subsequently, culture media was replaced with 180 µL of DMSO and cells incubated for 15 min to dissolve Formazan and absorbance (OD) was measured at 560 nm and evaluated by a SYNERGY-HT multiwell plate reader (Synergy-HT, Bio-Tek Instruments, Winooski, VT, USA). Each concentration was tested in triplicate and experiments were carried out twice.

In a preliminary experiment, H_2_O_2_ and Triton (0.05 and 0.1 Mm) were used to test the sensitivity of RAW264.7 cells to cytotoxicyty.

Viability was determined as a percentage of viable cells using the following the equation [54]:Cell Viability (%) = [(OD_S_ − OD_N_)/(OD_P_ − OD_N_)] × 100
where OD_S_ represents the absorbance of treated wells (RPMI + BNP/L2), OD_P_ represents the absorbance of non-treated wells (RPMI only) and OD_N_ represents the no-cells wells (RPMI only; background).

### 5.5. ELISA Assay for Inflammatory Cytokine Evaluation

RAW264.7 cells were seeded into 96-well plates at a density of 1 × 10^4^ cells/well and allowed to adhere for 24 h at 37 °C, then exposed overnight to LPS (1 µg/mL, L2630, Sigma-Aldrich Co., USA) to induce inflammation and cell activation. Different concentrations of L2 or BNP (0, 0.05, 0.1, 0.2, 0.4, 0.8, 1.6, 3.2 and 6.4 ng/mL) were added and incubated for 24 h to determine TNF-α and IL-6, and 48 h to assess IL-10. To determine whether the effects on inflammatory cytokines were mediated by natriuretic receptors, the cells were pretreated with isatin (0.1 mM), a L2/BNP antagonist [12], 20 min before L2 and BNP treatments. Cytokine quantification was performed on cell culture supernatant using, respectively, anti-Mouse TNF-α, anti-Mouse IL-6 and anti-Mouse IL-10 DuoSet ELISA kits (R&D Systems, Minneapolis, MN, USA) according to the manufacturer’s protocol. Briefly, plates were pre-coated with 100 µL of primary cytokines antibody (capture antibody) reconstituted in PBS following the manufacturer concentration, sealed and incubated overnight at room temperature (RT°). The following day, plates were washed 3 times with 400 µL of PBS containing 0.05% of Tween 20 using a ROCKER BioWasher200 Microplate ELISA Washer (ROCKER Scientific, Taiwan). Then, 300 µL of reagent diluent composed of 1% BSA in PBS was added to each well to saturate/block the plate and incubated for 1 h at RT° followed by a wash. Subsequently, 100 µL of 2× serial dilutions of cytokines standards reconstituted and diluted in reagent diluent, following manufacturer instructions, and diluted samples were added to assigned wells and incubated for 2 h at RT° followed by a wash. Following this, 100 µL of biotinylated cytokines detection antibody reconstituted and diluted in reagent diluent was added to each well and incubated for 2 h at RT° followed by a wash. Then, 100 µL of 40-fold-diluted streptavidin-HRP (Horse Radish Peroxidase) was added to each well and incubated for 20 min at RT° protected from light followed by a wash. Subsequently, 100 µL of TMB (Tetramethylbenzidine) and H_2_O_2_ 1:1 mixture substrate solution (TMB ELISA Substrate (High Sensitivity), Abcam, Cambridge, UK) was added to each well and incubated for 20 min at RT° protected from light, followed by the addition of 50 µL of 2N sulfuric acid (H_2_SO_4_) stop solution to each plate, and the optical density of each well was determined immediately using a TECAN SPARK multiplate reader (Tecan Group Ltd., Männedorf, Switzerland) at 450 nm wavelength with 540 nm correction. Each concentration was tested in triplicate and experiments were carried out twice.

### 5.6. Flow Cytometry for Macrophage Polarization Assessment

RAW264.7 macrophage polarization was detected after treatment with L2 and BNP by flow cytometric profiling of specific surface marker expression, including CD68 for total macrophage population quantification and CD206/MRC-1 for M2-like macrophage quantification using PE Rat Anti-Mouse CD68 and Alexa Fluor^®^ 647 Rat Anti-Mouse CD206 (BD BioScience, San Jose, CA, USA). Pro-inflammatory M1 were identified as CD68+ and CD206- cells, and anti-inflammatory M2 were identified as CD68+ and CD206+ cells. Cells were seeded in 24-well plates at a density of 5 × 10^4^ cells/well and inflammation was induced by exposure to LPS as described on ELISA protocol above. Three sets of experiments were carried out. In the first one, cells were treated with or without L2 or BNP (0.2, 0.4 and 0.8 ng/mL) and incubated for 48 h and pretreated with or without isatin (0.1 mM). In the second set, cells were treated with IL-10 at 10 and 20 ng/mL (R&D Systems, Minneapolis, MN, USA) and incubated for 48 h. In the third set, cells were pre-exposed to a 50-fold-diluted IL-10 antibody (10 µg/mL) in RPMI (Mouse IL-10 antibody, R&D Systems, Minneapolis, MN, USA) for 20 min then treated with L2 at 0.4 ng/mL for 48 h. Cells were harvested by trypsinazation (250 µL of Trypsin-EDTA) and collected in separate FACS tubes. Cell pellets were washed twice with 1 mL of PBS-FBS 1% following centrifugation at 1300 rpm for 10 min at 37 °C and the supernatant was discarded. After the elimination of excess buffer using filter paper, cells were re-suspended in a fixation/permeabilization solution (Fisher Scientific, Thermo Fisher Scientific, UK) for 20 min at 4 °C. Cells were then pelleted by centrifugation and washed twice with 1× Perm/wash supplemented with FBS and Saponin (Fisher Scientific, Thermo Fisher Scientific, UK). Pellets were resuspended in 300 µL of PBS-FBS 1%, then 1.25 µL of each PE Rat Anti-Mouse CD68 and Alexa Fluor^®^ 647 Rat Anti-Mouse CD206 antibodies (1:240, BD BioScience) was added to the tubes and incubated on ice for 30 min protected from light. Following two washes with PBS-FBS 1%, cells were re-suspended in 300 µL of Cell-Fix solution (Fisher Scientific, Thermo Fisher Scientific, UK). Tubes were then introduced into the BD FACSCanto™ Flow-cytometer (BD Biosciences) and cell populations were sorted and analyzed using BD CellQuestPro software (version 2.0, system OS2; Becton, Dickinson and Company). To ensure the accuracy of the experimental results, all samples were analyzed within 3 h to avoid fluorescence changes, which could affect the experimental results. The number of detected cells per tube was 8000–20,000.

### 5.7. Statistical Analysis

Results are expressed as mean ± SEM. Data analysis was performed using GraphPad Prism 8.0.2 for Windows, GraphPad Software (San Diego, CA, USA). Intergroup comparisons of mean values were performed by one-way analysis of variance followed by Student’s t test for further evaluation of differences between 2 means. Statistical significance was defined as *p* < 0.05.

## Figures and Tables

**Figure 1 toxins-15-00298-f001:**
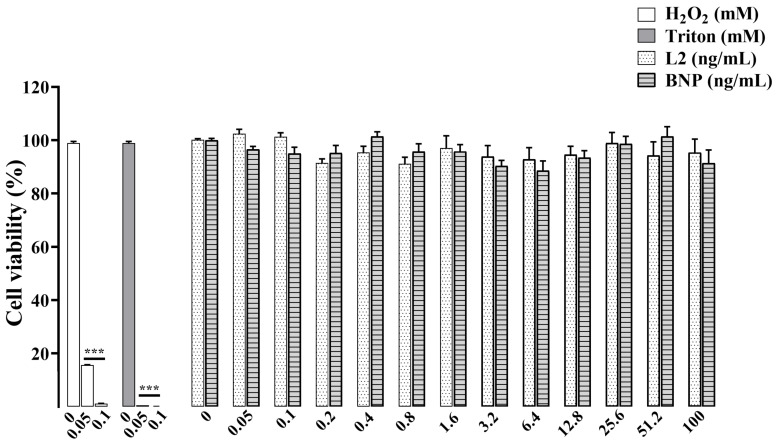
Cell viability by MTT assay on RAW 264.7 cells. Cells were exposed for 24 h to H_2_O_2_ or Triton (0.05 and 0.1 mM), and to L2 or BNP (0.05, 0.1, 0.2, 0.4, 0.8, 1.6, 3.2, 6.4, 12.8, 25.6, 51.2 and 100 ng/mL). n = 8–9/condition. Data are reported as mean ± SEM. *** *p* < 0.001 vs. corresponding untreated control group.

**Figure 2 toxins-15-00298-f002:**
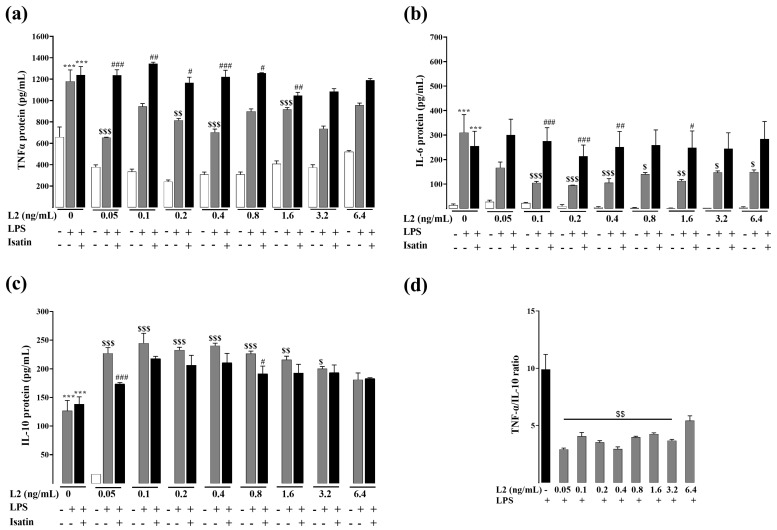
Effects of L2 on LPS-induced inflammatory cytokines by ELISA assay. Levels of (**a**) TNF-α, (**b**) IL-6 and (**c**) IL-10, and (**d**) TNF-α/IL-10 ratio after L2 treatment of LPS-activated RAW 264.7 cells. After LPS (1 µg/mL) priming for 24 h, cells were treated with different concentrations of L2 (0, 0.05, 0.1, 0.2, 0.4, 0.8, 1.6, 3.2 and 6.4 ng/mL) for 24 h to determine TNF-α and IL-6, and for 48 h to assess IL-10 levels. Natriuretic peptide receptor antagonist isatin (0.1 mM) was added 20 min before L2 treatment. n = 3–6/condition. Data are reported as mean ± SEM. *** *p* < 0.001 vs. corresponding control LPS untreated group; $ *p* < 0.05, $$ *p* < 0.01, $$$ *p* < 0.001 vs. corresponding control L2 untreated group; # *p* < 0.05, ## *p* < 0.01, ### *p* < 0.001 vs. corresponding LPS + L2 treated group.

**Figure 3 toxins-15-00298-f003:**
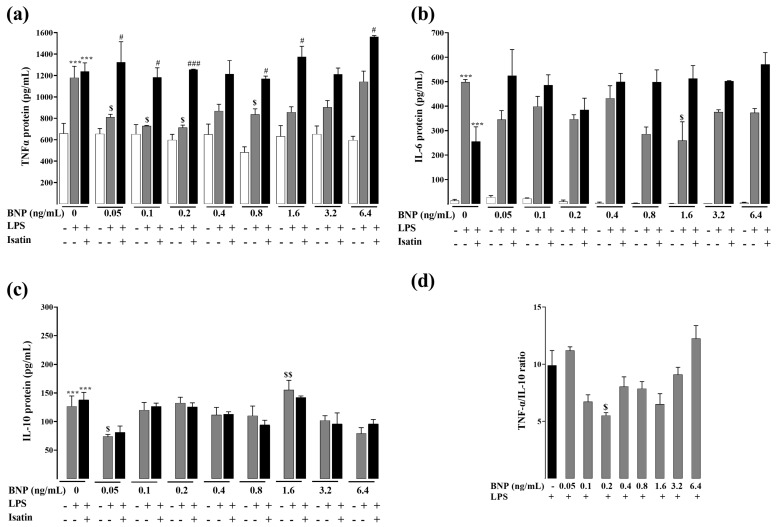
Effects of BNP on LPS-induced inflammatory cytokines by ELISA assay. Levels of (**a**) TNF-α, (**b**) IL-6 and (**c**) IL-10, and (**d**) TNF-α/IL-10 ratio after BNP treatment of LPS-activated RAW 264.7 cells. After LPS (1 µg/mL) priming for 24 h, cells were treated with different concentrations of BNP (0, 0.05, 0.1, 0.2, 0.4, 0.8, 1.6, 3.2 and 6.4 ng/mL) for 24 h to determine TNF-α and IL-6, and for 48 h to assess IL-10 levels. Natriuretic peptide receptor antagonist isatin (0.1 mM) was added 20 min before BNP treatment. n = 3–6/condition. Data are reported as mean ± SEM. *** *p* < 0.001 vs. corresponding control LPS untreated group; $ *p* < 0.05, $$ *p* < 0.01 vs. corresponding control BNP untreated group; # *p* < 0.05, ### *p* < 0.001 vs. corresponding LPS + BNP treated group.

**Figure 4 toxins-15-00298-f004:**
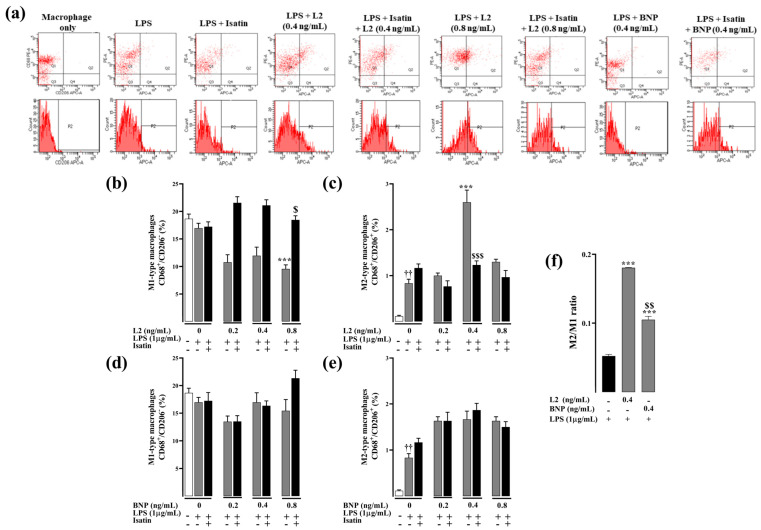
Effect of L2 and BNP on macrophage polarization in LPS-activated RAW264.7 cells. (**a**) Representative flow cytometry plots showing the variable macrophage distribution of M1-like macrophages (CD68^+^/CD206^−^ cells, Q1 population) and M2-like macrophages (CD68^+^/CD206^+^/MRC-1, Q2 population) in control and treated cells. (**b**,**c**) Effect of L2 on M1 and M2 macrophage subtype expression. (**d**,**e**) Effect of BNP on M1 and M2 macrophage subtype expression. Cells were obtained after challenging RAW264.7 cells with LPS (1 µg/mL) for 24 h followed by 48 h treatment with or without L2 or BNP (0, 0.2, 0.4 and 0.8 ng/mL) and isatin (0.1 mM), added 20 min earlier. (**f**) M2/M1 ratio in LPS-activated RAW264.7 cells. The subtypes of macrophages were identified by analyzing profiles of cell surface markers by FACS. Pro-inflammatory M1 macrophages were identified as CD68^+^/CD206^ࢤ^ cells and M2-like macrophages assessed by double immunolabeling of CD68 and CD206/MRC-1 in control and treated cells, and the data were analyzed by BD CellQuestPro software. All results were obtained from duplicate experiments. Data are reported as mean ± SEM. *** *p* < 0.001 vs. corresponding control LPS group; $ *p* < 0.05, $$ *p* < 0.01, $$$ *p* < 0.001 vs. L2 corresponding group; †† *p* < 0.01 vs. group unstimulated with LPS.

**Figure 5 toxins-15-00298-f005:**
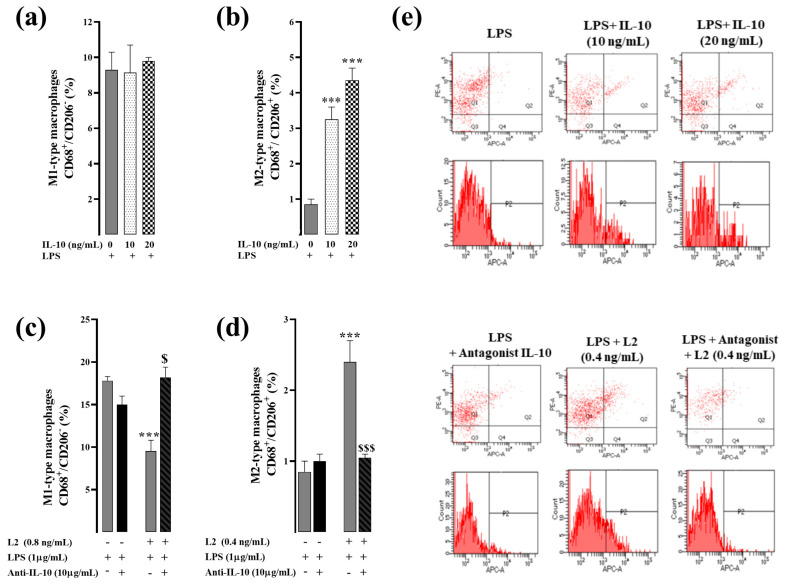
Effect of L2 on macrophage polarization in LPS-activated RAW264.7 cells after interleukin-10 inhibition. (**a**,**b**) Effect of interlekin-10 (IL-10) on M1 and M2 macrophage subtype expression. Cells were obtained after challenging RAW264.7 cells with LPS for 24 h followed by 48 h treatment with or without exogenous IL-10 (0, 10 and 20 ng/mL). (**c**,**d**) Effect of L2 on M1 and M2 macrophage subtype expression after IL-10 inhibition. Cells were obtained after challenging RAW264.7 cells with LPS (1 µg/mL) for 24 h followed by 48 h treatment with or without L2 (0, 0.4 or 0.8 ng/mL) and IL-10 inhibitor at 10 µg/mL, added 20 min earlier. (**e**) Representative flow cytometry plots showing the variable macrophage distribution of M1-like macrophages (CD68^+^/CD206^−^ cells, Q1 population) and M2-like macrophages (CD68^+^/CD206^+^/MRC-1, Q2 population) in control and treated cells. The subtypes of macrophages were identified by analyzing profiles of cell surface markers by FACS. Pro-inflammatory M1 macrophages were identified as CD68^+^/CD206^−^ cells and M2-like macrophages assessed by double immunolabeling of CD68 and CD206/MRC-1 in control and treated cells, and the data were analyzed by BD CellQuestPro software. All results were obtained from duplicate experiments. Data are reported as mean ± SEM. *** *p* < 0.001 vs. corresponding control LPS group; $ *p* < 0.05, $$$ *p* < 0.001 vs. L2 corresponding group.

**Figure 6 toxins-15-00298-f006:**
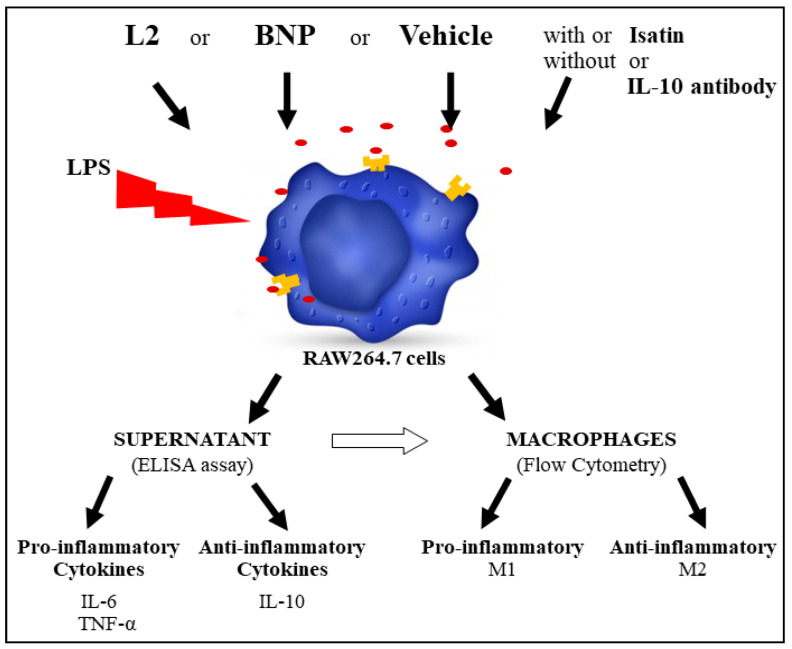
Protocol to investigate the effect of L2 on macrophage polarization in vitro on lipopolysaccharide (LPS)-activated RAW264.7 macrophages and to explore the associated underlying mechanisms.

## Data Availability

All data generated or analyzed during this study are included in this manuscript.

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
