# Peer review of "Natriuretic-like Peptide Lebetin 2 Mediates M2 Macrophage Polarization in LPS-Activated RAW264.7 Cells in an IL-10-Dependent Manner"

_toxins, 2023, doi:10.3390/toxins15040298_

Round 1
Reviewer 1 Report
In the manuscript entitled “Natriuretic like peptide Lebetin 2 mediates M2 macrophage polarization in LPS activated RAW264.7 cells in an IL 10 dependent manner” Authors took an effort to determine the effect of the snake natriuretic like peptide L2 on activation of macrophages. The study present results from in vitro cell-based experiments using mice macrophages RAW 264.7. The manuscript contains potentially interesting and important data noteworthy for the readers; it is well written and easy to follow; the methodology is also quite well written. However, in my opinion, the manuscript requires some modifications within the presentation of obtained results. Additionally, low number of experiments (n=2) does not allow to obtain reliable conclusions and require experiments repetition before resubmission.
If possible, the answers and comments should be included in a new version of manuscript, since they will allow simpler following of presented data.
-Please explain, why concentration 6.4 ng/mL of peptides was chosen for further studies, since L2 at 0.4 ng/mL already decreased the metabolic activity by more than 15%.
-Figure 1 – I suggest to present the effect of H2O2 and Triton on metabolic activity (the positive control) on the same scheme as for L2 and BNP (1a is unnecessary). Results presented in Figure 1 were obtained from n=2/3 repeats; in my opinion the experiments should be performed with higher number of repeats to obtain reliable data.
-In my opinion further data (following the peptides effect on metabolic activity of RAW cells) should be presented in other way using as a control the cells treated with LPS only. Of course, Authors should also present the data obtained for cells without treatment with LPS. What is more, on the Figure 2 there is not presented effect of only LPS treatment on secretion of TNFa, IL-6 and Il-10 – this should be presented as additional bar in Figure 2a (for TNFa), 2b (for Il-6), 2c (for Il1-0). Figure 2d does not present data in proper way and should be removed.
-The same comment is for Figure 3, which does not present effects of only LPS treatment on secretion of TNFa, IL-6 and Il-10 – this should be presented as additional bar in Figure 3a (for TNFa), 3b (for Il-6), 3c (for Il1-0). Figure 3d does not present data in proper way and should be removed.
- In Figure 4 bars presenting the effect of LPS on cells should be added to corresponding schemes. Why the concentration 0.8 mg/mL was studied (it was already cytotoxic for RAW cells), whereas in previous experiments 6.4 mg/mL was used (also cytotoxic). Please, present the results for cells unstimulated with LPS.
- Results presented in Figures 4 and 5 were obtained from duplicate experiments; in my opinion the experiments should be performed with higher number of repeats to obtain reliable data.
-The Authors already work with RAW cells and I suggest to add results presenting the effect of L2/BNP on the NO secretion after cells stimulation by LPS (i.e. with Griess reagent).
-Please, compare the structures of BNP and L2, and discuss the differences of the responses.
-in methodology add the dilution of antibodies used in flow cytometry; what was the solvent for peptides – water? please confirm that for cells seeding or staining cell were harvested by trypsinazation not scrapped.
In summary, the manuscript requires the major revision and some experiments should be repeated before its resubmission.
Reviewer 2 Report
This paper entitled “Natriuretic-like peptide Lebetin 2 mediates M2 macrophage polarization in LPS-activated RAW264.7 cells in an IL-10 dependent manner” reports on the beneficial effects of snake natriuretic peptide (NP) Lebetin 2 (L2) in reducing inflammation and promoting M2-type macrophages in a reperfused myocardial infarction (MI) model. The study aimed to investigate the underlying mechanisms of L2 in macrophage polarization in lipopolysaccharide (LPS)-activated RAW264.7 cells in vitro. It provides valuable insights into the mechanism of action of L2 in promoting M2 macrophage polarization and reducing inflammation. These findings may have important implications for the development of new therapeutic strategies for cardiovascular diseases and other conditions associated with chronic inflammation. However, there few question should be addressed.
1. IL-4 and Arg-1 are also the indicators of M2 polarization, why did the author just detect IL-10?
2. This study only used a single cell line (RAW264.7 cells) to investigate the effects of L2 on macrophage polarization. While RAW264.7 cells are commonly used as a model for macrophage studies, it would be important to validate these findings in primary macrophages and in vivo models.
3. “in vivo” or “in vitro” should be in italics.
Author Response
Response to Reviewer 2 Comments
Point 1: This paper entitled “Natriuretic-like peptide Lebetin 2 mediates M2 macrophage polarization in LPS-activated RAW264.7 cells in an IL-10 dependent manner” reports on the beneficial effects of snake natriuretic peptide (NP) Lebetin 2 (L2) in reducing inflammation and promoting M2-type macrophages in a reperfused myocardial infarction (MI) model. The study aimed to investigate the underlying mechanisms of L2 in macrophage polarization in lipopolysaccharide (LPS)-activated RAW264.7 cells in vitro. It provides valuable insights into the mechanism of action of L2 in promoting M2 macrophage polarization and reducing inflammation. These findings may have important implications for the development of new therapeutic strategies for cardiovascular diseases and other conditions associated with chronic inflammation. However, there few question should be addressed.
Point 1. IL-4 and Arg-1 are also the indicators of M2 polarization, why did the author just detect IL-10?
Response 1: First, we thank the reviewer for the time spent reading and editing this article, and for this very useful comment. We agree that in addition to IL-10, other factors such as IL-4, TGF-β and ARG-1 could trigger the M1/M2 macrophage polarization (Orecchioni, M., 2019, Front immunol). However, in this work, we focused on the effect of L2 on IL-10 (and not other M1/M2 polarization factors such as IL-4 or ARG-1) as we found that, compared to BNP, only L2 increased the IL-10 levels in a sustained and consistent manner. Furthermore, in the literature, IL-10 has been reported as the major determinant of M1/M2 macrophage conversion (Chuang, Y., 2016, Innate Immun), particularly in the context of myocardial ischemia (Weirather, J., 2014, Circ Res ; Liu, Y., 2022, Biomater Res ; Nahrendorf, M., 2010, Circulation ; Tsujita, K., 2007, Circulation ; Shiraishi, M., 2016, J Clin Invest ; Dayan, V., 2011, Basic Res Cardiol) ; and that only a sustained and constant increase in IL-10 levels can trigger the M1/M2 switch (Lopes, R. L, 2016, Cytokine ; Chuang, Y., 2016, Innate Immun). Overall, these data prompted us to explore the role of IL-10 in the M2 polarizing effect of L2. In our next studies, as recommended by the reviewer, we will deeply investigate the anti-inflammatory aspects of L2 by exploring its effect on a wider panel of inflammatory factors including IL-4 and TGF-β, using the LEGENDplex™ HU Essential Immune Response Panel (13-plex) w/FPThis assay panel, with higher detection sensitivity and broader dynamic range than traditional ELISA methods. Regarding ARG-1, its overall level and activity will be assessed in cell culture supernatant using an end-point colorimetric method to evaluate the ornithine production, using L-arginine as a substrate (Canè S., et al., 2020, Methods Enzymol). We will also determine the involvement of these factors in the L2-induced M1/M2 macrophage polarization. However, these experiments cannot be performed to be presented in the present work and will therefore be published later.
Point 2: This study only used a single cell line (RAW264.7 cells) to investigate the effects of L2 on macrophage polarization. While RAW264.7 cells are commonly used as a model for macrophage studies, it would be important to validate these findings in primary macrophages and in vivo models.
Response 2: We thank the reviewer for this constructive comment and agree that it is important to validate our current data in other in vitro or in vivo models. We previously demonstrated that L2 exerts anti-inflammatory effects in vivo, in a rat model of myocardial ischemia-reperfusion, by reducing leukocyte and macrophage infiltration in the infarcted area (Tourki, B., et al., 2019, Toxins). Specifically, we found that only L2 and not BNP induces M2 polarization in vivo, which is consistent with the current in vitro findings. However, in addition to these results, our in vitro studies have allowed us a better mechanistic understanding of the inflammatory actions of L2, as well as a deeper investigation of the pathaway activated by L2 during M1/M2 conversion.
Regarding in vitro models for macrophage studies, here we used RAW264.7 cells, a murine macrophage-like cell line, which closely mimics bone marrow-derived macrophages in terms of cell surface receptors and response to ligands that initiate inflammatory activation (Berghaus L.J., 2010, Comp Immunol Microbiol Infect Dis). However, several studies reported advantages and drawbacks of using either cell models (i.e. primary macrophage-lineage cells versus RAW264.7 macrophages) (Berghaus L.J., 2010, Comp Immunol Microbiol Infect Dis ; Heideveld, E., 2020, Methods in Enzymology). For example, caution should be applied when extrapolating findings obtained on RAW264.7 cells, as their phenotype and function may change with continuous culture. On the other hand, primary macrophage cultures are more likely to be heterogeneous, and isolation of these cells can lead to their activation and differentiation (Heideveld, E., 2020, Methods in Enzymology). Therefore, in our future studies, we will consider using different cell models, as cross-checking the complementary information provided by each cell culture type could allow for more reliable data and conclusions.
Point 3: “in vivo” or “in vitro” should be in italics.
Response 3: We agree with the reviewer's suggestion and have changed the words "in vivo" and "in vitro" to italics in the revised version of the manuscript (in the original manuscript, "in vitro" and "in vivo" were written in roman type as we noticed that these words are usually edited and published in roman type in the journals of MDPI group).
Review Report Form
Open Review
(x) I would not like to sign my review report
( ) I would like to sign my review report
Quality of English Language
( ) English very difficult to understand/incomprehensible
( ) Extensive editing of English language and style required
( ) Moderate English changes required
(x) English language and style are fine/minor spell check required
( ) I am not qualified to assess the quality of English in this paper
|
Yes |
Can be improved |
Must be improved |
Not applicable |
|
|
Does the introduction provide sufficient background and include all relevant references? |
(x) |
( ) |
( ) |
( ) |
|
Are all the cited references relevant to the research? |
( ) |
( ) |
( ) |
( ) |
|
Is the research design appropriate? |
( ) |
(x) |
( ) |
( ) |
|
Are the methods adequately described? |
(x) |
( ) |
( ) |
( ) |
|
Are the results clearly presented? |
(x) |
( ) |
( ) |
( ) |
|
Are the conclusions supported by the results? |
(x) |
( ) |
( ) |
( ) |
As requested by the reviewer, English language and style were reviewed and a spell check was performed.
The research design was also improved by adding a diagram summarizing the general experimental protocol used in this work (5. Materials and Methods Section/ 5.2. Experimental research design_ in the revised manuscript).

Round 2
Reviewer 1 Report
I have read the answers and the submitted manuscript: the Authors answered some of my concerns and manuscript has been improved even by addition of results from new repeated experiments. However, in the Authors’ answer the graphs presenting results from additionally repeated experiments (Figures 4 and 5 for n = 2) are the same as graphs presenting the final results (Figures 4 and 5 for n = 4). I think that this issue requires the explanation.
